# Applying a Virtual Art Therapy System Based on the Michelangelo Effect in Patients with Spinal Cord Injury

**DOI:** 10.3390/s25134173

**Published:** 2025-07-04

**Authors:** Michela Franzò, Sara De Angelis, Marco Iosa, Gaetano Tieri, Giorgia Corsini, Giovanni Generoso Cellupica, Valentina Loi, Fabiano Bini, Franco Marinozzi, Giorgio Scivoletto, Federica Tamburella

**Affiliations:** 1Department of Psychology, Sapienza University of Rome, 00185 Rome, Italy; michela.franzo@uniroma1.it (M.F.); corsini.1787049@studenti.uniroma1.it (G.C.); cellupica.1848744@studenti.uniroma1.it (G.G.C.); 2IRCCS Fondazione Santa Lucia, 00179 Rome, Italy; s.deangelis@hsantalucia.it (S.D.A.); gaetano.tieri@unitelmasapienza.it (G.T.); valentinaloi0701@gmail.com (V.L.); g.scivoletto@hsantalucia.it (G.S.); f.tamburella@unilink.it (F.T.); 3Virtual Reality & Digital Neuroscience Lab, Department of Law and Digital Society, Unitelma Sapienza University, 00161 Rome, Italy; 4Department of Mechanical and Aerospace Engineering, Sapienza University of Rome, 00185 Rome, Italy; fabiano.bini@uniroma1.it (F.B.); franco.marinozzi@uniroma1.it (F.M.); 5Department of Life Sciences, Health and Health Professions, Link Campus University of Rome, 00165 Rome, Italy

**Keywords:** neuroaesthetic, virtual reality, neurorehabilitation, art therapy

## Abstract

**Highlights:**

**What are the main findings?**
High scores in terms of USEQ and NASA were reported for the proposed Virtual Art Therapy System both for patients and healthy subjects. It can be administered in patients with spinal cord injury for the rehabilitation of their upper limbs.Most of the kinematic variables automatically measured by the system were significantly different between patients and healthy subjects. Analysis in the frequency domain of subjects’ movements showed a high horizontal (but not vertical) variability in the spectrum.

**What is the implication of the main finding?**
Virtual Art Therapy System is a comfortable and user-friendly technique for administering upper-limb therapy in patients with spinal cord injury.The System also provides quantitative measures of the kinematic parameters of patients that are helpful in assessing patients’ deficits and performances.The spectral analysis could be helpful for verifying the match between the rehabilitation aims and the task executions.

**Abstract:**

Background: Serious videogames have already demonstrated their positive impact on rehabilitation and of particular interest is the virtual reality (VR) technology. This immersive technology has been used in this study to create a neuroaesthetic experience based on the Michelangelo effect for the rehabilitation of patients with spinal cord injury. The aim of this study was to test the usability of a system for virtual art therapy and its capacity to assess patients’ deficits performances. Methods: A VR headset was worn by the participants who experienced a painting simulation of famous artworks (artistic stimuli) against a coloring canvas (non-artistic stimuli). The trajectories of the hand were studied to obtain different kinematic and spectral parameters to evaluate the user performances. A total of 13 healthy subjects and 13 patients with spinal cord injury participated in this study. Results: Significative differences were obtained for most of the parameters between the two groups, except for the normalized jerk and energy of the spectrum. Analysis in the frequency domain showed that both groups preferred horizontal movements for painting the canvas. The NASA and USEQ scores reported a comfortable and user-friendly system according to the patients’ point of view. Conclusions: The system can be a usable tool, the rehabilitative efficacy of which should be tested in patients with spinal cord injury. The kinematic and spectral parameters would allow for the evaluation of the performances alongside the clinical scales, distinguish pathological and physiological performances.

## 1. Introduction

Virtual reality (VR) is an innovative technology that allows users wearing a head-mounted display to immerse themselves in a different three-dimensional world with which they can interact with hand-controllers. VR is already used in several fields, and different suggestions and applications have been developed for the medical field as well. VR has been used in healthcare education [1,2,3] and in clinical practice, to alienate patients from stress and reclusion [4] or to integrate traditional rehabilitative protocols [5]. In the literature, different VR applications have been proposed as rehabilitative approaches for patients with neurological conditions [6,7,8], such as post-stroke and Alzheimer diseases. The videogames, with the aim of motivating fulfillment of the therapeutic requirements, improving physical fitness, and reducing symptoms of diseases, have been named “serious games”, and they have been demonstrated successful and a valuable option for elderly subjects as well [9]. These games can be customized to suit the user’s abilities, allowing progress to be tracked and constantly improved. Compared to traditional methods, the use of VR allows for greater motivation and involvement, improving the rehabilitation experience and, consequently, clinical outcomes [10]. Mixed reality has also been used in a similar context with positive results [11,12]: it is a holographic experience that complements the real world instead of an immersive digital experience.

Moreover, some studies have explored how VR can be combined with art therapy to treat anxiety and social difficulties [13] or to motivate patients during neurorehabilitation [14,15]. Art has recently been recognized to have positive effects on health promotion and disease treatment [16]. However, the art therapy protocols are rarely founded on neuroaesthetic principles, and the outcomes of art therapy are not often quantitatively investigated [16].

In [17], an aesthetic experience in a virtual environment was presented as an upper-limb rehabilitative application for patients with neurological disorders. The immersive, serious game was developed for VR headset, and it consisted of a painting activity. In front of the user, a white canvas was displayed, and the user would move his/her hand, grabbing the controller, to touch the canvas, using it as a virtual brush for painting. The patient has the illusion of being able to paint an art masterpiece because when the controller is in contact with the canvas it deletes the pixels of a white thin panel overlaying the already present picture of the artwork. With this experience, the patient can perform therapeutic movements with the arm while facing an engaging activity. The rehabilitative protocol has been performed by patients with stroke, and its efficacy has been demonstrated [18,19]. In these previous studies, both patients and healthy subjects reported to perceive less fatigue during the virtual reproduction of artistic stimuli than during the simple coloring of the canvas, and their performances resulted in a more accurate presence of artistic stimuli: these effects of the art were called the Michelangelo effect [18,19]. These VR systems also allow for the recording of several kinematic and quantitative data useful for a post-processing evaluation of the performance through the calculation of different parameters (i.e., time to complete the trial, percentage of artwork discovered, normalized jerk, length of the trajectory made by the hand, etc.). Despite also being potentially useful in other neurological diseases, this protocol of virtual art therapy has been applied only on patients with stroke until today.

In this study, this Virtual Art Therapy System was employed for the first time in patients with spinal cord injury (SCI). In addition to the above-mentioned kinematic parameters, a frequency analysis of patients’ upper-limb movements were also performed to assess the usability and the possible utility of this system in rehabilitation.

Global estimates suggest that in 2021, approximately 15.4 million people were living with a spinal cord injury (WHO Spinal cord injury fact sheet, 2024). In Western Europe in particular, recent studies [20] documented an incidence of SCI between 16 and 19.4 new cases per million inhabitants per year. Spinal cord injuries represent a complex condition that has a significant impact on the patient’s life, making the rehabilitation process crucial for the recovery of impaired motor and cognitive functions. The mostly used technologies for rehabilitation trainings in patients with SCI are transcranial magnetic stimulation, functional electrical stimulation, and robotic-assisted therapy [21]. However, rehabilitation aims not only to improve physical abilities but also to support the patient’s psychological and social well-being, with the goal of restoring, as far as is possible, autonomy in daily activities. Moreover, the deliverability and home-caring aspects should also be considered [22]. The effectiveness of the treatment depends on a personalized approach that combines traditional rehabilitation techniques, such as the repetitive execution of task-oriented exercises, and more modern methodologies [23]. In addition to the motor part, spinal cord injury rehabilitation must also address cognitive impairments, which are frequently present. Cognitive deficits, such as memory loss and difficulty in attention and executive processes, further limit autonomy and quality of life. However, traditional cognitive rehabilitation techniques usually focus on single, specific cognitive domains, without addressing the complexity of cognitive functions involved in everyday activities in an integrated manner. For this reason, it is essential to adopt an approach that considers the entire spectrum of cognitive abilities, promoting global rather than isolated improvement of individual skills and motivating approaches to rehabilitation [24].

Several reviews [25,26,27,28,29,30] have collected and analyzed the literature findings to evaluate and assess the applicability of VR protocols in patients with SCI. They conclude in considering VR as a promising technology to produce interventions for SCI patients’ rehabilitation approaches and that it can be used also for pain management [31]. Also, mixed reality has been used for the same purposes [32,33,34].

The aim of this study is to make a first evaluation of the Virtual Art Therapy System for patients with spinal cord injury, testing its usability and performing an innovative statistical analysis of the kinematic data in order to assess new quantitative parameters, which may be used in the future for randomized clinical trials aiming at testing the efficacy of this system. Obviously, we hypothesized a lower performance for patients with respect to that of healthy subjects, but we also hypothesized that these differences could be reduced in the presence of artistic stimuli leading to the higher engagement of patients, in accordance with the Michelangelo effect.

## 2. Materials and Methods

### 2.1. Virtual Art Therapy System: Hardware and Software

The system is composed of a head-mounted display (HMD) for VR experience, the Meta Quest 2 with one controller (Meta, Menlo Park, CA, USA). The serious game was developed in Unity engine (version 2018) built and installed on the HMD to make it work as a stand-alone device. The data about the user performances are saved automatically as a txt file in the device memory (frequency of acquisition: 50 Hz), and they can be downloaded by connecting the HMD to the computer. The System was previously tested in healthy subjects and patients with stroke [17,18,19] but never in patients with spinal cord injury. Each subject sat wearing the HMD and used the controller in his/hand as a virtual brush for painting on a white canvas in the virtual environment, as described in previous studies [17,18] and briefly detailed in the following sections.

### 2.2. Protocol of Acquisition

The painting experience is developed to be performed with one hand using only one controller. The other controller can be used by the therapist to manage the experience, such as changing the canvas or positioning the position of the subject in the virtual environment with respect to the canvas to personalize the space according to the user’s arm length and motor capability. The participant wearing the HMD moved the controller grasped in his/her hand to interact with the canvas and to paint it (Figure 1). The subject could interact with the canvas with a virtual sphere, displayed in the VR environment in the same position of the oculus controller in the real environment with respect to the HMD. Each virtual canvas appeared white at the beginning of the task. Subjects were instructed that the sphere acts as a brush, coloring the canvas when put in contact with it, forming a painting that could be an artistic masterpiece or a mix of colors forming abstract figures (control stimuli). The illusion of painting is given thanks to a white thin virtual panel placed in front of the canvas which occluded the visibility of an underlying image. When the subject touched the virtual panel with the brush, the target pixels were automatically deleted, allowing the participant to see a part of the underlying picture of a famous painting or a control image. Control images were obtained by blur-filtered reversed (both left–right and up–down) version of paintings to maintain the same palette of colors and the same amount of brightness in the artistic masterpieces (Figure 1). The participant has no guide or indications about how to move the hand during painting. The system was able to record the x,y,z coordinates of the hand movements.

The protocol consists in a first calibration phase, in which the user set the canvas in the correct position in space and the system records the capability of the user to reach the corners of the canvas in both the horizontal and vertical orientations. After the calibration, two different lists of 30 randomized stimuli each were presented to the user. All the canvases had a rectangular shape (60 cm × 40 cm); half of the stimuli were oriented vertically and the other half horizontally. The two lists referred to artistic and non-artistic stimuli. The first list of stimuli was formed by high-resolution pictures of famous paintings (such as Starry Night of van Gogh or the Creation of Adam of Michelangelo). The second list was formed by a set of undefinable stimuli maintaining the same color palette and the same amount of brightness in the art masterpieces, these stimuli were obtained by a blur-filtering of the original paintings with a low resolution to make the feature unrecognizable (according to the control stimuli previously used [17]). The order of submission of the lists to the participants has been randomized. Each experimental session lasted about twenty minutes. During this time, the participant was required to uncover as many canvases as possible. The time of each session also included some breaks, which were necessary to allow the participant to rest.

The protocol of acquisition is common to both groups of subjects. The main difference concerned the presence of the physiotherapist close to the patient during the experience. Considering the system as a tool for clinicians, the physiotherapist had the liberty to intervene to help the patient in the activity.

To evaluate the usability of the system, after the experiment, the NASA Task Load Index [35] and the User Satisfaction Evaluation Questionnaire (USEQ) [36] were evaluated for both the artistic and non-artistic stimuli lists and both the groups of participants. USEQ has 6 questions testing the self-perceived satisfaction, efficacy, efficiency, easiness-to-use, fatigue, and utility of the performed exercise, with a five-point Likert Scale for each one of these items with a score ranging from 1 to 5, and hence a total score ranging from 6 (poor satisfaction) to 30 (excellent satisfaction). NASA-TLX has six questions testing self-perceived mental demand, physical demand, time demand, effort, performance, and stress, with a ten-point numerical rating scale for each one of these items.

### 2.3. Participants

Experiments on the group of 13 healthy subjects (30 ± 7 years old) were conducted in the laboratory of Industrial Bioengineering of Sapienza University of Rome, they presented no comorbidities or musculoskeletal diseases, and they were volunteers providing informed consent before the experiment. Because the aim of enrolling a control group was to obtain the reference values of the best possible performance during the task, we enrolled young adults.

The group of 13 patients (50 ± 20 years old) with spinal cord injury consisted of patients admitted to the IRCCS Fondazione Santa Lucia, where the experimental activities took place. This study was approved by the Lazio Area 5 Territorial Ethics Committee. Before the training session, clinical staff evaluated the patient with different clinical scales: Spinal Cord Independence Measure (SCIM); Graded Redefined Assessment of Strength, Sensibility, and Prehension (GRASSP); Upper Extremity Motor Score (UEMS); Modified Ashworth Scale (MAS).

The inclusion criteria, against which patients were selected for virtual reality rehabilitation training, were mainly focused on the functional level of patients. Inclusion criteria were as follows: cognitive and motor abilities sufficient enough to understand and perform the required task, injury occurred at cervical level, (patients classified according to the International Standards for Neurological Classification of Spinal Cord Injury (ISNCSCI) with impairment grade C or D as defined by the scale proposed by the American Spinal Injury Association (ASIA), and UEMS score higher than 12. The exclusion criteria were focused on severe comorbidities: respiratory non-autonomous subjects, with severe neurological nonspecific spinal cord injury associated with upper-limb impairment (e.g., brachial plexus injury, severe cerebral plexus injury, etc.), as well as subjects with a medical history of visual disturbances or a diagnosis of photosensitive epilepsy or with seizure episodes. Given the strict inclusion–exclusion criteria, no other restrictions were applied to the enrolment of subjects in terms of age or etiology of spinal cord injury events (both traumatic and non-traumatic injuries were included).

Information of the patients are reported in Table 1.

### 2.4. Kinematic Data Acquisition and Processing

The data saved during the performance are as follows:Controller’s coordinates in space and time;“score”: the percentage of canvas pixels discovered;These data were elaborated and analyzed in post-processing with a MATLAB (version 24.2.0.2871072, R2024b, Update 5, MathWorks Inc., Natick, MA, USA) specific script. About the controller’s trajectories, only the *x* and *y* coordinates were considered to study the plane of the canvas; then, the trajectories have been elaborated selecting only the samples related to the painting activity and applying a moving average filter. From these data, some parameters have been calculated, first of all, the kinematic quantities:Time to complete the trial (s);Normalized jerk (NJ), calculated as in [37];Length of the trajectory covered by the hand (m), calculated as the pathway performed on the frontal plane in which the canvas was laid;Patients’ performances are expected to be characterized by a longer time needed to complete each trial and a longer trajectory with respect to healthy subjects. The jerk is the derivative of the acceleration with respect to time and, after being normalized for trial duration, it is a measure of roughness and irregularity in the brush trajectory [37]. It is expected to be higher in patients than in healthy subjects, because in the latter people the movements are usually more smoothed and fluid.Then, spectral quantities were computed by an analysis in the frequency domain for the trajectory of, separately, the x and y components, taking into account the sampling frequency of the system (50 Hz). The following parameters were computed:Dominant frequency of the power spectrum (Hz), selected as the frequency of the spectrum with the highest magnitude;Mean value of magnitude of the spectrum (in meters, m);Energy spectrum (m^2^), calculated as the sum of the squares of the amplitudes of the spectrum;Variance of the spectrum (m^2^), calculated as the dispersion of the spectral content around its centroid.

The movements of healthy subjects are expected to be characterized by higher frequencies, more energy and larger variance on the spectrum than patients.

### 2.5. Statistical Analysis

Each parameter has been calculated for both artistic and non-artistic stimuli and for each participant. Then, medians and quartiles (shown in box–whiskers plot) and means and standard deviations (reported in tables) for each group have been computed. An inferential statistical mixed analysis of variance was performed to compare the results between the two groups of subjects, using the type of stimulus (artistic vs. non-artistic) as a within-subject factor. This analysis is generally robust for a non-normal data distribution and was used because it allows combining between- and within-subject effects. However, since most of the data did not pass a normality check conducted using the Shapiro–Wilk test and Mann–Whitney u-test was also applied to compare patients’ and healthy subjects’ variables for artistic and non-artistic stimuli. For all the analyses the alpha level of statistical significance was set at 0.05.

## 3. Results

### 3.1. Kinematic and Spectral Quantities

Figure 2 and Figure 3 show the results about the values of the parameters calculated for both groups, healthy subjects and patients, and for both lists, non-artistic (NS) and artistic stimuli (AS). All subjects completed the painting of the canvas, so the score was 100% for all of them and not reported in the Figures.

The time to complete the trial was significantly different between groups (F(1,24) = 19.3, *p* < 0.001, η_p_^2^ = 0.449), with longer durations for patients, but it did not depend on the type of stimulus (*p* = 0.872), nor on the stimulus*group interaction (*p* = 0.986). A similar result was obtained for the length of the trajectory, which was significantly affected only by a main effect of group (F(1,24) = 16.6, *p* < 0.001, η_p_^2^ = 0.408). Conversely, normalized jerk depended only on the stimulus*group interaction (F(1,24) = 4.526, *p* = 0.05, η_p_^2^ = 0.156), with artistic stimuli reducing jerk in patients.

The dominant frequency on the x-axis depended only on the group (F(1,24) = 25.2, *p* < 0.001, η_p_^2^ = 0.512), but not on the stimulus (F(1,24) = 0.265, *p* = 0.611, η_p_^2^ = 0.011), nor on the stimulus*group interaction (F(1,24) = 0.812, *p* = 0.376, η_p_^2^ = 0.033). A significant group effect was also observed for all other variables extracted from spectral analysis (*p* < 0.05). The main effect of stimulus approached statistical significance for spectral energy along the vertical axis (F(1,24) = 3.43, *p* = 0.076, η_p_^2^ = 0.125), which was also partially influenced by the stimulus*group interaction (F(1,24) = 3.76, *p* = 0.064, η_p_^2^ = 0.135), with higher spectral energy values observed in patients for artistic stimuli.

Table 2 and Table 3 report the results (*p*-values) of the non-parametric Mann–Whitney u-tests for each parameter between the two groups of subjects for both non-artistic and artistic stimuli.

### 3.2. Usability

To evaluate the usability of the system, NASA and USEQ questionnaires were submitted to the participants and the final scores were calculated. The USEQ item mean value was 4.24 (on 5) for non-artistic stimuli and 4.37 (on 5) for artistic paintings with a final comprehensive score of 25.46 and 26.23, respectively. Moreover, 4.69 and 5 were the scores for the “clarity” item and 4.69 and 4.46 for the “successful use”. No statistically significant differences were found between healthy subjects and patients.

While the mean values for the NASA among the items were for non-artistic and artistic stimuli, respectively, reported on the bar chart of Figure 4. The *t*-test applied to compare the two lists showed only a significantly different score for the “Temporal Demand” item. Despite a longer time being required for artistic paintings, neither effort nor frustration were significantly higher for these stimuli.

## 4. Discussion

The main objective of this research was to verify the usability and feasibility of an innovative approach to upper-limb rehabilitation based on art therapy administered by an immersive virtual reality system for patients with spinal cord injury.

The analyses conducted showed significant and promising results that provided a quantitative perspective in the use of art therapy to support rehabilitation methods. Statistical analysis of the parameters in comparison with the control group of healthy subjects showed several significant results in both categories of stimuli. Among the kinematic quantities, the time to complete the trial and the length of the trajectory were significantly longer in patients, as expected. The most interesting result was obtained for normalized jerk, which was not significantly affected by the main effect of group. However, the difference between healthy subjects and patients with spinal cord injury was statistically significant only for the interaction between group and type of the stimulus, with artistic stimuli reducing normalized jerk in patients. Previous studies have shown that normalized jerk may depend on the perceived beauty of the image [38], and that lower values are associated with more fluid movements during a virtual sculpturing task [39]. In our study, the main effect of group was not statistically significant, but the interaction between group and stimulus was. Non-parametric tests partially confirmed these results, with a *p*-value close to the significant threshold for non-artistic stimuli (*p* = 0.069) and far from it for artistic stimuli (*p* = 0.4). For normalized jerk, as well as for time to complete the trial and hand trajectory length, the differences between patients and healthy subjects were less pronounced in the presence of artistic stimuli (i.e., higher *p*-values), suggesting an effect of artistic stimuli in improving the performance of patients.

Another innovative aspect of this study was the spectral analysis that enabled the obtainment of parameters from the frequency analysis of the one-dimensional trajectories over time of the subject’s hand that were not calculated in previous applications of this system [17,18,19], but it was already used in previous studies on patients with spinal cord injury to investigate hand trajectories [40] and standing ability [41]. The results of our study demonstrate that both healthy and pathological subjects, in front of a canvas and without guides to follow, prioritize the horizontal movements of the hand over the vertical ones, as demonstrated by the fact that the dominant frequencies on the y-axis are zero. Furthermore, the other spectral parameters, the energy, variance, and amplitude of the spectrum, were significantly lower in patients than in healthy subjects. This result could be associated with the main deficit in shoulder flexion of patients with spinal cord injury and to the need for help in countering gravity during the upper-limb treatment of these patients. In further studies, this analysis could be important to verify the match between the rehabilitative aims and the actual execution of the tasks by the patients. Again, the presence of artistic stimuli may improve patients’ performance: the energy spectrum of horizontal hand movements was significantly lower in patients than in healthy subjects for non-artistic stimuli, but not for artistic paintings. A mixed ANOVA partially confirmed these results along the vertical axis as well, showing a trend toward the statistically significant threshold, with higher spectral energy for patients in the presence of artistic stimuli. These findings are consistent with the Michelangelo effect [17].

Despite some less-evident results, such as those related to normalized jerk, the quantitative analysis of kinematic parameters can be used to assess the patient’s performance improvements during rehabilitation. This, in turn, can support the physiotherapists in selecting and adapting individualized rehabilitation protocols. It will be particularly supportive to further confirm the validity of these quantitative parameters when correlations with qualitative clinical scales are made. In this way, assessment scales can be created to complement the clinical ones.

Furthermore, it is interesting to note that although the patients were in some cases helped by physiotherapists who deemed it necessary for the rehabilitation pathway of the patients, the results also showed a clear difference with the reference group.

Virtual reality was already used in the rehabilitation of patients with spinal cord injury for the recovery of walking ability [26], pain relief [30], and upper-limb functional recovery [42]. Also, the art therapy was administered on these patients, but painting protocols were mainly used only for improving their mood and well-being [43,44]. Our study was the first one that combined these two approaches in patients with spinal cord injury.

The System was based on the Michelangelo effect that was previously described as an effect of art that reduces the fatigue and improves the performance of patients [17,18,19]. In the present study, it could be observed in some parameters recorded for patients that resulted in being more similar to those recorded for healthy subjects for artistic stimuli and not for control stimuli, such as the energy spectrum along the x-axis and the normalized jerk. Patients reported a higher temporal demand for artistic stimuli, despite not perceiving more fatigue during the execution of this task, and no significant differences were observed in terms of the time needed to complete the task.

It must be borne in mind that no inclusion criteria were set for this study regarding the age of the participants. In fact, the age range of the patients is very wide, with the only limitation being the age of majority, while for the control group, an attempt was made to create a reference group corresponding to the best possible performance.

A limit of our study is the sample size: the small sample of patients may not allow generalization of the results obtained and that the lack of long-term follow-up limits the possibility of assessing the sustainability of the observed improvements over time. These activities will be carried out in subsequent studies in order to suggest clinical practice guidelines based on evidence from a randomized controlled trial [45].

The results obtained from the NASA and USEQ scales by the patients support the usability of the system. In particular, the scores provided for the USEQ items demonstrate that the system was understood and appreciated by all the patients. The NASA results show that the total workload is 40%, but it is necessary to consider the wide range of age and degree of pathology of the patients. In fact, the minimum score for workload was 6.67, and the maximum was 78.33. Furthermore, no significant differences were found between the artistic and non-artistic stimuli, for both indexes. The only significant item was the perception of time required by the exercise, which was greater for the artistic stimuli than for non-artistic stimuli, despite it not affecting the fatigue perceived by patients. This can be explained by considering that for the artistic stimulus the subject was more inclined to observe the painting instead of proceeding quickly in the mere gesture of uncovering the canvas.

## 5. Conclusions

The immersive virtual system based on the Michelangelo effect was used on a group of patients with spinal cord injury and on a control group of healthy subjects. Kinematic and spectral parameters were calculated and compared between the two groups. The results obtained support the hypothesis that these parameters allow for quantitatively evaluating the performance of the patients and that therefore they could be useful for future acquisitions and the creation of quantitative scales to support purely qualitative clinical scales. The presence of artistic stimuli had an effect mainly on patients, improving the spectral energy and fluidity (as indicated by a reduction in normalized jerk) of their hand trajectories. Furthermore, the NASA and USEQ scales have received positive evaluation feedback for the system from patients, suggesting that it is easy to understand and use. These results should be considered as the combined effect of art therapy (which was proposed for patients with spinal cord injury more than 30 years ago [46] and whose efficacy was more recently endorsed by the World Health Organization [47]) and virtual reality, which simultaneously enabled the enjoyment of beautiful and motivational artistic masterpieces and the illusion of being able to reproduce them through virtual painting.

## Figures and Tables

**Figure 1 sensors-25-04173-f001:**
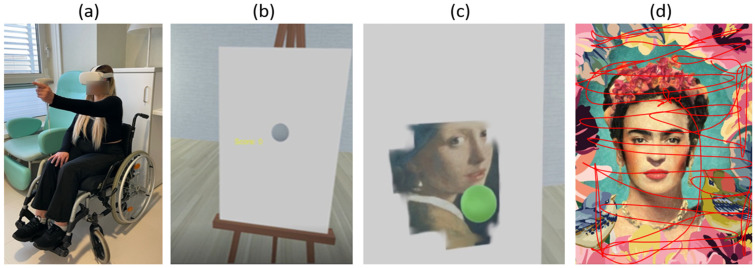
(**a**) Photo of the patient during the experiment; (**b**) the white canvas in the virtual environment; the grey sphere represents the virtual brush as explained to the patient; (**c**) example of an artistic masterpiece during the patient’s painting; the spherical brush became green when the brush was in contact with the canvas to provide feedback to the patient; (**d**) an example of an artistic stimulus with the superimposed hand trajectory recorded by the system and analyzed to obtain the kinematic parameters.

**Figure 2 sensors-25-04173-f002:**
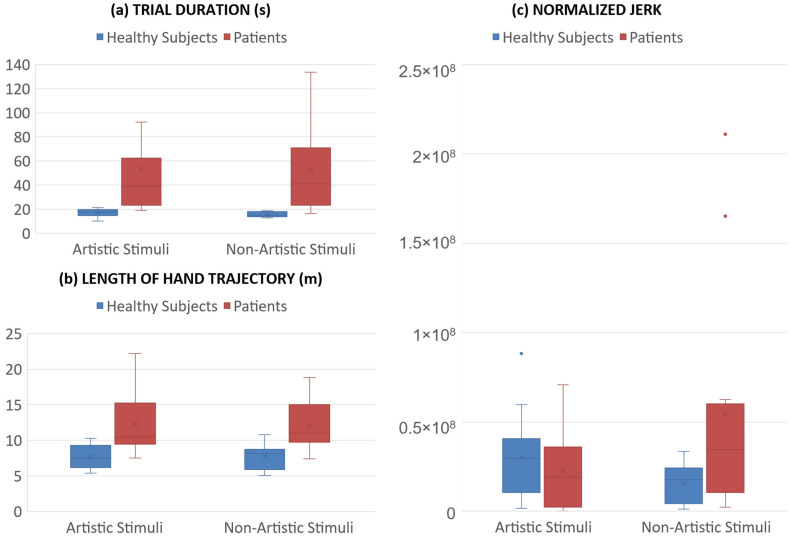
Boxplots of the kinematic quantities comparing the healthy subjects (blue) and patients (red) for both artistic (blue) and non-artistic (red) stimuli: (**a**) time duration of the trial; (**b**) length of the hand trajectory; (**c**) normalized jerk.

**Figure 3 sensors-25-04173-f003:**
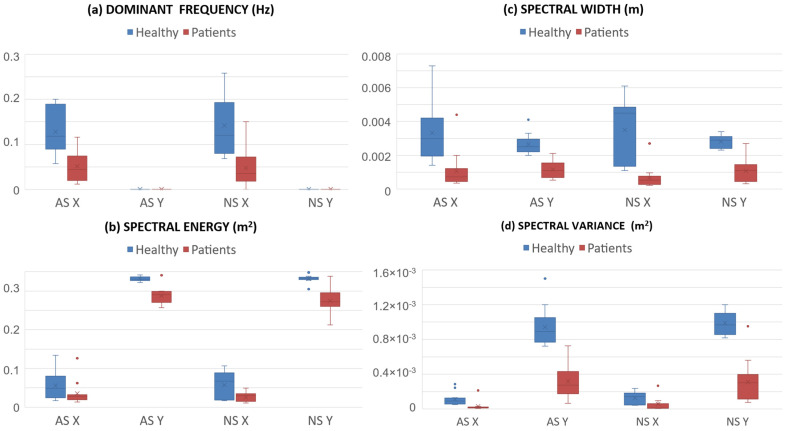
Boxplots of the spectrum comparing the healthy subjects (blue) and patients (red) for both artistic (AS) and non-artistic (NS) stimuli and for both horizontal (x) and vertical (y) axis: (**a**) dominant frequency; (**b**) spectral energy; (**c**) spectral width; (**d**) spectral variance.

**Figure 4 sensors-25-04173-f004:**
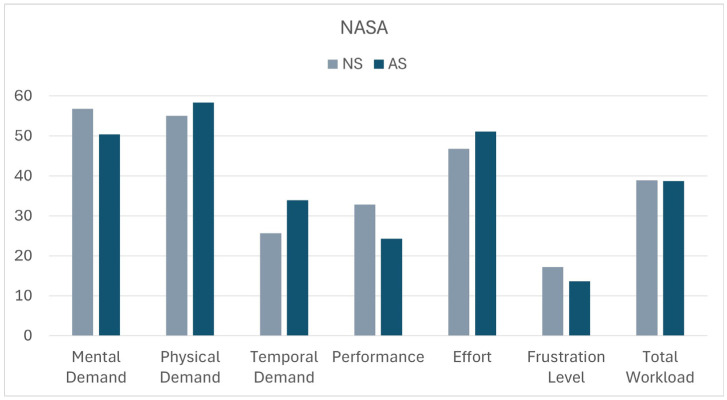
Bar chart summarizing the mean scores among the patients for each item of the NASA score for both non-artistic (light blue) and artistic (dark blue) stimuli lists. Items are rated within a 100-point range with 5-point steps.

**Table 1 sensors-25-04173-t001:** Clinical information on the patients (T: Traumatic; NT: not traumatic etiology; level of lesion; ASIA: American Spinal Injury Association scale; GRASSP: Graded Redefined Assessment of Strength, Sensibility, and Prehension; DH: dominant hand; NDH: non-dominant hand; UEMS: Upper Extremity Motor Score; MAS: Modified Ashworth Scale; SCIM: Spinal Cord Independence Measure).

Age	T/NT	LEVEL	ASIA	GRASSP DH	GRASSP NDH	UEMS	MAS	SCIM
68	T	C6	C	68	51	38	2	21
64	T	C5	C	63	66	36	5	63
65	T	C4	D	64	47	26	16	27
29	NT	C5	C	70	69	32	1	68
51	T	C5	C	72	36	28	6.5	21
48	T	C5	C	81	81	38	4.5	68
60	T	C5	C	20	27	18	8	13
55	NT	C1	D	55	59	34	5.5	38
66	T	C5	C	39	44	29	7.5	10
64	NT	C5	D	57	58	32	7.5	79
52	T	C5	D	92	91	48	1	79
66	T	C4	D	81	81	40	9	58
66	T	C3	D	59	62	31	7.5	76

**Table 2 sensors-25-04173-t002:** Means ± standard deviations (SDs) for the kinematic parameters computed on the Healthy Group (HG) and Patient Group (PG) with relevant *p*-values for the between-group comparisons (all the *p*-values refer to Wilcoxon test, * *p* < 0.05).

Kinematic Parameter	NON-ARTISTIC STIMULI			ARTISTIC PAINTINGS		
	HG Mean ± SD	PG Mean ± SD	*p*-Value	HG Mean ± SD	PG Mean ± SD	*p*-Value
**time to complete the trial (s)**	15.6 ± 2.0	52.2 ± 34.9	<0.001 *	17 ± 3	53.3 ± 46.6	<0.001 *
**normalized jerk (NJ)**	1.56 ± 1 × 10^7^	5.44 ± 6.6 × 10^7^	0.068	3.04 ± 2.4 × 10^7^	2.26 ± 2.3 × 10^7^	0.336
**length of hand trajectory (m)**	7.77 ± 1.74	12.10 ± 3.56	<0.001 *	7.67 ± 1.67	12.3 ± 4.68	<0.001 *

**Table 3 sensors-25-04173-t003:** Means ± standard deviations (SDs) for the spectral quantities computed on the Healthy Group (HG) and Patient Group (PG) with relevant *p*-values for the between-group comparisons (all the *p*-values refer to Wilcoxon test, * *p* < 0.05).

Spectral Parameter		NON-ARTISTIC STIMULI			ARTISTIC PAINTINGS		
		HG Mean ± SD	PG Mean ± SD	*p*-Value	HG Mean ± SD	PG Mean ± SD	*p*-Value
**dominant frequency of the power** **spectrum (Hz)**	**X**	0.14 ± 0.07	0.05 ± 0.04	<0.001 *	0.13 ± 0.05	0.05 ± 0.03	<0.001 *
	**Y**	0	0	-	0	0	-
**mean value of amplitude spectrum (m)**	**X**	35 ± 18 × 10^−4^	6.6 ± 6.5 × 10^−4^	<0.001 *	33.2 ± 16 × 10^−4^	10.6 ± 11 × 10^−4^	<0.001 *
	**Y**	28.2 ± 4.0 × 10^−4^	10.7 ± 7.5 × 10^−4^	<0.001 *	26.5 ± 5.8 × 10^−4^	11.5 ± 5.4 × 10^−4^	<0.001 *
**energy spectrum (m^2^)**	**X**	0.057 ± 0.035	0.026 ± 0.012	0.022 *	0.065 ± 0.034	0.035 ± 0.03	0.058
	**Y**	0.332 ± 0.096	0.274 ± 0.029	<0.001 *	0.332 ± 0.006	0.288 ± 0.022	<0.001
**variance of the** **spectrum (m^2^)**	**X**	1.2 ± 0.7 × 10^−4^	5.8 ± 9.8 × 10^−5^	0.005 *	1.1 ± 0.7 × 10^−4^	3.0 ± 5.6 × 10^−5^	<0.001 *
	**Y**	9.9 ± 1.3 × 10^−4^	3.1 ± 2.5 × 10^−4^	<0.001 *	9.4 ± 2.2 × 10^−4^	3.2 ± 1.8 × 10^−4^	<0.001 *

## Data Availability

Because the data involve clinical information of patients, an anonymous database is available only after motivated requests to the corresponding author.

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
