# Peer review of "Applying a Virtual Art Therapy System Based on the Michelangelo Effect in Patients with Spinal Cord Injury"

_sensors, 2025, doi:10.3390/s25134173_

Round 1
Reviewer 1 Report
Comments and Suggestions for Authors
The manuscript has inconsistent and confusing terminologies, such as in the case of the two subject groups (healthy and patients) -- there is no clear definition of what 'control' and 'experiment' groups are, but they were later used in the discussion section. Furthermore, the 'control' and 'experiment' terminologies have already been used in describing two different stimuli.
The manuscript also has several inadequate explanations or details -- such as what exactly the tasks are that participants are supposed to do in the experiment?, how to interpret the outcomes of some of the measurements, such as jerk, spectrum?, what are the actual USEQ questions and what is the sample size of the control (healthy) group? On the main concept of the Michelangelo Effect, a detailed explanation is required earlier in the paper to establish the framework on which this entire study is on.
From the two points above, it makes this manuscript difficult to follow, understand its content and evaluate the significance of its findings.
Even with that lack of clarity, I managed to spot some issues in the results and discussion sections. There is no reporting of the normality test. The statistical tests only tested if there is a difference between the patient and healthy groups (a null hypothesis), but the test should indeed indicate the direction of the difference ( alternative hypothesis -- e.g. healthy group is better than the patient group). Some of the discussions are not well supported, such as "reduction of fatigue" -where is the data reported in the results and the min/max values of the workload - they were not shown in the results but got discussed in the discussion. And lastly, the comparative tests were only between healthy and patient groups (which I am still unclear what exact insight that can provide) -- what is about the comparison between 'Control' and 'Experimental' stimuli? -- even though it is not very clear what the purpose of having the two stimulus types is.

The manuscript requires significant work in correcting grammatical errors and improving the clarity of some of the sentences. Please refer to the attached document with highlights showing some issues I was able to spot. Please ensure that a comprehensive proofreading is conducted for the revision.
Reviewer 2 Report
Comments and Suggestions for Authors
This study evaluates the usability + feasibility of a virtual therapy approach for upper limb rehabilitation in patients with injury of the spinal cord. The approach leverages. VR technology combined with neurasthenic principles (i.e., Michelangelo Effect), where participants exposed to experimental stimuli (i.e., famous paintings) or control stimuli (i.e., abstract blurred images) using hand movements.
Overall, this research is very interesting and a good read. However, there are minor flaws that either need a more elaborate clarification or a better justification of choices as the following:
- Authors should be clear from the onset that there is evidence of reduced fatigue or improved performance from being exposed to art stimuli, which in contradictory to prior studies in stroke patients.
- I have concerns about recruiting the participants with a wide range of age (i.e., 50 ± 20 years) and also high variability in lesion levels (i.e., C1-C6), which could confound the results. Authors should elaborate more about the choices that have been made during the inclusion criteria.
- While there were no significant differences in normalized jerk and energy spectrum; what was the purpose of utilize them as a reliable measure.
Round 2
Reviewer 1 Report
Comments and Suggestions for Authors
The changes in the revision significantly improve the overall quality of the paper.
Author Response
We would like to thank the Reviewer for the positive judgement about the new version of our manuscript, and for the qualified suggestions provided in the previous round that helped us in improving our manuscript